# Employment of Neuromuscular Electrical Stimulation to Examine Muscle and Bone Qualities after Spinal Cord Injury

**DOI:** 10.3390/jcm11226681

**Published:** 2022-11-11

**Authors:** Ashraf S. Gorgey, Refka E. Khalil, Tommy W. Sutor, Jacob A. Goldsmith, David X. Cifu

**Affiliations:** 1Spinal Cord Injury and Disorders Hunter Holmes McGuire VA Medical Center, Richmond, VA 23249, USA; 2Department of Physical Medicine and Rehabilitation, Virginia Commonwealth University, Richmond, VA 23284, USA

**Keywords:** spinal cord injury, neuromuscular electrical stimulation, amplitude of the current, muscle quality, bone quality, bone mineral density

## Abstract

(1) Background: Resource intensive imaging tools have been employed to examine muscle and bone qualities after spinal cord injury (SCI). We tested the hypothesis that surface neuromuscular electrical stimulation (NMES) amplitude can be used to examine knee extensor muscle quality, distal femur and proximal tibia bone mineral density (BMD) in persons with SCI. (2) Methods: Seventeen persons (2 women) with chronic SCI participated in three weeks of NMES-resistance training twice weekly of 4 sets of 10 repetitions. Participants were classified according to the current amplitude (>100 mA) and the number of repetitions (>70 reps) of leg extension into greater (*n* = 8; 1 woman; group A) and lower (*n* = 9; 1 woman; group B) musculoskeletal qualities. Magnetic resonance imaging, dual energy x-ray absorptiometry, isometric peak torque, Modified Ashworth and Penn spasm frequency scales were conducted. (3) Results: In between group comparisons, current amplitude was lower (38–46%) in group A. Whole (27–32%; *p* = 0.02), absolute (26–33%, *p* = 0.02) thigh muscle and absolute knee extensor muscle cross-sectional areas (22–33%, *p* = 0.04) were greater in group A. Right distal femur (24%; *p* = 0.08) and proximal tibia (29%; *p* = 0.03) BMDs were lower in group B, and peak isometric torque (*p* < 0.01), extensor spasticity scorers (*p* = 0.04) and muscle spasm scores (*p* = 0.002) were significantly higher in group A. Regression models revealed that amplitude of current, repetitions and body weight can accurately predict musculoskeletal qualities in persons with SCI. (4) Conclusions: Surface NMES amplitude and repetitions of leg extension differentiated between SCI survivors with greater versus lower musculoskeletal qualities. The study may shed the light on the interplay between muscle and bone in persons with SCI.

## 1. Introduction

Clinical trials are underway exploring the potential of restoring over ground ambulation in persons with spinal cord injury (SCI) [1,2]. Neuromodulation techniques are gaining interest in the field of SCI to restore standing, stepping and overground ambulation [2,3]. These trials have demonstrated that long-term, task-specific training using either body weight supported treadmill training or exoskeleton can assist walking effectively by engaging the lumbosacral neuro circuitries to enhance neuroplasticity and restore mobility [3,4]. These trials rely primarily on common principles of enhancing activity-dependent plasticity by ensuring frequent repetitions and high intensity training [3]. As these interventions are likely to impose significant stressors on muscle and bone below the level of injury, ensuring that persons with SCI have adequate muscle and bone qualities should be considered as predetermining criteria for participation in any of these trials to maximize safety.

SCI has long-term and deleterious effects on muscle and bone qualities. Within few weeks post-injury, loss in skeletal muscle cross-sectional area (CSA) may reach to 50% of the original pre-injury size [5,6]. Furthermore, skeletal muscle atrophy is accompanied with infiltration of intramuscular fat (IMF) [7,8,9]. This depot of ectopic adipose tissue not only exerts metabolic dysfunction at the muscle cell, but also creates a mechanical disadvantage by reducing muscle stiffness [10]. A recent study employing magnetic resonance elastography showed a 12% decrease in knee extensor stiffness in persons with SCI compared to healthy able-bodied controls [10]. The decrease in stiffness of the vastus lateralis muscle was clearly accentuated with increased IMF in persons with SCI and this decrease in stiffness may interfere with muscle quality and likely lead to detrimental effects on bone quality [11,12]. Persons with chronic SCI also experience progressive bone demineralization that leads to neurogenic osteoporosis [13,14]. Following SCI, osteoporosis is characterized by decreasing bone mineral density (BMD) at a rate of 2% to 4% per month in the first year after SCI, reaching a steady state within 2 to 3 years post SCI [15,16,17], but may be extended up to 8 years [18]. Persons with SCI have double the risk of fracturing their lower extremities compared to healthy able-bodied [19,20,21]. The distal femur and proximal tibia experience approximately 50% of total BMD loss within 2 to 3 years post SCI due to mechanical unloading [19], and not surprisingly, fractures most frequently occur at the distal femur and the proximal tibia after experiencing neurogenic osteoporosis [13].

Resource intensive imaging techniques are currently employed to evaluate muscle quality and bone health in persons with SCI. Musculoskeletal quality is defined as enhanced peak torque, greater absolute muscle CSA (i.e., without IMF) and BMD in persons with SCI. The use of MRI is considered the gold-standard in evaluating muscle adaptations in response to unloading, disuse, SCI or following a period of training [5,7,11], and magnetic resonance imaging (MRI) has been used to evaluate muscle quality and trabecular bone changes following surface neuromuscular electrical stimulation-resistance training (NMES-RT) [11,12]. Despite the risk of ionization, computerized tomography has also been used to adequately evaluate volumetric changes in the trabecular bone at the distal femur or proximal tibia [22]. Dual energy x-ray absorptiometry (DXA) has also been used to evaluate BMD at the most common sites of neurogenic osteoporosis in persons with SCI [13]. Recently, our laboratory has employed regional ilunar DXA to evaluate the effects of long-term intrathecal baclofen on BMD and to determine the effectiveness of using total body scan to evaluate knee BMD at both the distal femur and proximal tibia [23].

Today, most clinicians do not have access to many of these costly and high-technology imaging tools and, thus, to rapid and effective way to examine those who are at risks of experiencing poor muscle quality and osteoporosis in persons with SCI. In a pilot trial [24], current amplitude of surface NMES that was applied to the knee extensor muscle group successfully differentiated among persons with SCI with different muscle qualities. Those with greater muscle quality (i.e., greater muscle CSA and lower IMF) required less current amplitude (<100 mA) to evoke leg extension [24]. In this investigation, we propose the use of surface NMES as an effective diagnostic tool to determine those who are at risk of developing reduced musculoskeletal quality, and hypothesize that using an arbitrary value of 100 mA [24] along with the number of leg extension repetitions may allow for the determination of knee extensor muscle quality and distal femur and proximal tibia BMD. We further hypothesize that those individuals with enhanced muscle quality will have greater muscle CSA, lower %IMF and greater peak torque of the knee extensor muscle group and greater BMD at the distal femur and proximal tibia.

## 2. Materials and Methods

### 2.1. Participants

Participants were recruited from a registered clinical trial NCT02660073 [25]. The study protocol was approved by Institutional Review Boards at Hunter Holmes McGuire Veterans Affairs Medical Center and Virginia Commonwealth University. Following a written informed consent, each participant underwent a physical examination performed by a study physician. Seventeen individuals, with chronic (>12 months post injury) SCI (C5-L1; International Standards for Neurological Classification of Spinal Cord Injury Classification A or B) were included. The demographics and physical characteristics of both groups are presented in Table 1. We have adopted both cross-sectional (two groups with different musculoskeletal qualities; A and B) and repeated measure (over 3 weeks) designs to address the primary research questions. Participants were then classified according to their response to the current amplitude (>100 mA) and the number of repetitions (>70 reps) of leg extension into greater (*n* = 8; 1 woman; group A) and lower (*n* = 9; 1 woman; group B) musculoskeletal quality groups. Detailed study inclusion and exclusion criteria were previously published [25].

### 2.2. Interventions

Neuromuscular electrical stimulation resistance training (NMES-RT) was the targeted intervention used. A Theratouch 4.7 stimulator unit (Rich-Mar, Inola, OK, USA) was manually adjusted at a current amplitude (mA) sufficient to evoke full leg extension against gravity [26].

NMES-RT sessions were approximately 45–60 min long [11,26]. The goal was completion of four sets of 10 knee extensions on each leg. While seated in their wheelchair, participants were first instructed to remove their shoes. A pillow placed behind the participant’s legs provided cushioning to prevent the legs from hitting the wheelchair. NMES was delivered via two surface electrodes placed on the skin over the knee extensor muscles of each leg. The distal electrode was placed medially over the vastus medialis, approximately one-third the distance from the patella to the inguinal fold. The proximal electrode was placed laterally, adjacent to the inguinal fold over the vastus lateralis. A rectangular, biphasic waveform was delivered (450 μs pulse duration with a 50 μs inter-pulse interval) at 30 Hz. The current amplitude was increased gradually until full knee extension was achieved. This methodolgoy was prevoiulsy used to meaure muscle fatigue non-invasively in realtime using mechanomyography in persons with SCI and was called maximum electrically stimulated extension [27]. After achieving full knee extension, the current amplitude for each repetition was recorded.

Leg extension was maintained for 3–5 s to evoke maximum tension in the activated muscle fibers. The current amplitude was then gradually decreased, returning the leg to its starting position [27]. 3–5 s of rest was provided between each repetition and ~2–3 min between sets. The first two sessions (in week one) were performed without ankle weights to ensure that participants could extend their lower legs against gravity. Ankle weights were added beginning in week two if the participant achieved full knee extension. This approach was employed to ensure full knee extension [26]. The current amplitude necessary to evoke a full knee extension for the first three weeks (2 sessions per week) was recorded. With participants seated in their personal wheelchair, a successful repetition was determined if the lower leg was capable of moving against gravity from 90° to 5° or less (i.e., 0° is full knee extension). The amplitude of the current (mA) of the first (1–10 reps) and second (11–20 reps) sets were averaged separately from the amplitude of the current of the third (21–30 reps) and fourth (31–40 reps) sets. This strategy was adopted because knee extensor muscle fatigue commonly ensues by the beginning of the third set of training and requires greater current amplitude to evoke full leg extension. In this study, a current amplitude of less than 100 mA and total number of knee extension repetitions (>70 out of 80 reps) were used as criteria to classify participants into group A (greater musculoskeletal quality) and group B (lower musculoskeletal quality).

### 2.3. Measurements: Dual Energy X-ray Absorptiometry (DXA)

Body composition was measured by whole body scans using a General Electric (GE) Lunar Prodigy Advance scanner (GE Lunar Inc., Madison, WI, USA). Lean mass for legs was measured by DXA [28]. The DXA scanner was calibrated using a daily quality control phantom according to manufacturer guidelines. Participants were transferred to the DXA table using either a ceiling lift or self-transfer with or without a sliding board. Participants were allowed 20 min in a flat supine position to account for possible fluid shifts before starting the scan. Knees were strapped together using a large Velcro strap above the knee joints and every effort was made to ensure that each leg was placed in neutral position with the big toe facing upward. Whole-body posture was aligned straight with no rotation in the pelvis or shifting of the trunk. All scans were performed and analyzed by a trained DXA operator using Lunar software version 10.5. Total body cuts were placed by the computer auto analysis program delineating anatomical regions of interest and final adjustments were made to ensure optimum inter-participant reproducibility. Short-and long-term precision of the regional and whole-body composition using DXA was previously determined in persons with SCI [28].

Knee BMD values were computed at proximal tibia and distal femur using the manufacturer software Lunar EnCore version 16 (GE Healthcare, Madison, WI, USA) [23]. For the proximal tibia, a rectangular region of interest was drawn with its height set at 7% of femur length, width set to include only tibial bone (excluding fibula) and its proximal edge positioned at the uppermost point of contact between the fibular head and the tibia. For the distal femur, a rectangular region of interest was drawn with its height set to match the proximal tibia metaphysis, width to include femoral bone and its lateral end positioned at a distance of 13% of femoral length extended up from the lateral condyle. Manual correction, using a brushing tool within the software was used to assign bone pixels that were not automatically detected by the software. Previous work showed that in chronic SCI group, the root-mean-square coefficient of variation (RMS-CV) values for BMD were 3.12 and 3.40%, for the distal femur epiphysis and proximal tibia epiphysis, respectively [23].

### 2.4. Magnetic Resonance Imaging (MRI)

MRI was captured from the hip joint to the knee joint (thigh) using a whole-body coil [26,27,29,30]. A fast spin-echo sequence was completed using a 1.5-Tesla magnet (GE Signa, Waukesha, WI, USA) and localized GE body array flex coil to provide an adequate image resolution and signal-to-noise ratio (repetition time, 850–1000 ms; echo time, 6.7 ms; field of view, 20 cm; matrix, 256 × 256). CSA of thigh muscle groups was derived from trans-axial images (0.8 cm thick, 1.6 cm apart) captured from the femoral head to the knee joint.

Images were analyzed with Win-Vessel (Ronald Meyer, Michigan State University, East Lansing, MI, USA). Images were automatically segmented into background/bone, skeletal muscle, and fat (low, medium, and high intensity, respectively). Intensity variations caused by radio frequency heterogeneity were corrected with a first pass segmentation. A fuzzy c-mean clustering algorithm was subsequently used to re-segment the corrected image to the three intensity components. In each selected image, CSA (cm^2^) was quantified by manually tracing anatomical regions of interest pixel-by-pixel.

To distinguish muscle from fat, the outer perimeter of the thigh muscle group was manually traced, and pixel signal intensity was automatically determined via the software. A bimodal histogram segmentation was plotted that contained two distinct peaks, with the first peak representing the threshold for muscle and the second peak representing the threshold for fat. This mid-point value was used to separate muscle pixels from IMF pixels as previously described [29,30]. Absolute skeletal muscle was determined via signal intensity after excluding IMF and femoral bone CSA.

### 2.5. Modified Ashworth Scale and Penn Spasms Frequency Scale

Modified Ashworth Scale (MAS) was used to evaluate spasticity for hip, knee, and ankle skeletal muscle flexors and extensors for both lower extremities in a blinded fashion as previously described [31,32]. Spasticity was evaluated in a supine position by the same by a study physician who was completely blinded to the design of the study (Table 2). The room temperature was held constant at 21 °C to 24 °C. Spasticity was measured between 12:00–2:00 PM for all the study participants. The Penn spasms frequency scale was also self-reported for the study participants. The scale is composed of 5 categorical points including 0; no spasms, 1; mild spasms occurring less than once per hour, 2; infrequent full spasms occurring less once per hour, 3; spasms occurring more than once per hour and 4; spasms occurring more than 10 times per hour [33].

### 2.6. Isometric Peak Torque

Participants were transferred to the Biodex chair using a ceiling lift and were secured using shoulder straps in a seated position. The trunk-thigh and thigh-leg angles were set to 85° and 90°, respectively [11]. The dynamometer was aligned to the anatomical knee axis, and the lever arm of the knee attachment was positioned 2–3 cm superior to the lateral malleolus. At least 2 complete isometric knee extensor contractions per limb and were elicited via NMES at a frequency of 10 Hz, 30 Hz and 100 Hz using the same device, stimulation parameters, and electrode placement as described above. A constant 100 mA current intensity was provided during testing for all participants using a biphasic pulse duration of 450 µs. The data were sampled at 1000 Hz and processed by a single trained researcher using LabChart (Version 7.3.8; ADInstuments, Colorado Springs, CO, USA). Data from isometric actions (2 trials) were averaged for each limb. The onset and cessation of isometric torque activity were visually identified by an evaluator blinded to subjects’ group assignment. Isometric peak torque was calculated by computing the average of a ~600 ms window during the signal plateau quickly following the rise of the torque signal.

### 2.7. Statistical Analysis

All data were tested for normality using Shapiro–Wilk tests. Data that were not normally distributed were log transformed prior to analyses. Outliers were detected using normal Q-Q plots. The means ± standard deviations (SD) of each outcome variable were then calculated, and independent t-tests were conducted to analyze the differences between group A (*n* = 8) and B (*n* = 9). Pearson correlations were conducted to determine the agreement in the amplitude of the current (mA) across the three weeks of the study. Linear regression analyses were conducted to establish models that predict muscle CSA after using the amplitude of the current (mA) as an independent variable. Statistical analyses were performed using IBM-SPSS version 28.0 (SPSS, Chicago, IL, USA). Statistical significance was set at an alpha level of *p* ≤ 0.05.

## 3. Results

Participant demographics, physical measurements and injury characteristics are presented in Table 1. All the tested variables were normally distributed without any detected outliers.

### 3.1. NMES-Amplitude and Repetitions

The amplitudes of the current (mA) required to evoke full knee extension for the first three weeks are listed in Table 2. The amplitude of the current (mA) in week 1 was positively related to that of weeks 2 and 3 for both the right (r = 0.92; *p* < 0.001) and left (r = 0.79–0.95; *p* < 0.001) legs. Based on 100 mA cut-off, the average amplitude of the current (mA) was 44–46%, 39–46%) and 47–54% lower for group A in weeks 1, 2 and 3, respectively, compared to group B (Table 2). Week 1 of the NMES-RT was conducted without ankle weights (0 lbs.). For week 2 [right: 2 ± 0 vs. 0.2 ± 0.7 lbs; *p* < 0.0001 and left: 1.8 ± 0.7 vs. 0.28 ± 0.76 lbs; *p* < 0.0004] and week 3 [right: 4 ± 0 vs. 0.4 ± 0.8 lbs; *p* < 0.0001 and left: 3.2 ± 1.7 vs. 0.86 ± 1 lbs; *p* < 0.0002] group A lifted greater ankle weights compared to group B (Table 2).

Group A achieved more leg extension repetitions for the right (40 reps total per session) and left (40 reps total) legs. The total number of repetitions out of 80 in week 1 [right: 75 ± 14 vs. 50 ± 22 reps; *p* = 0.01 and left: 75 ± 13 vs. 55 ± 21 reps; *p* = 0.04] and week 2 [right: 80 ± 0 vs. 56 ± 17 reps; *p* = 0.002 and left: 80 ± 0 vs. 64 ± 24 reps; *p* = 0.07] confirmed this finding, but not in week 3 after adding additional ankle weights compared to group B.

Liner regression analyses were conducted after using amplitude of the current, knee extension repetitions and body weight as independent variables to predict muscle CSAs, IMF CSAs and BMD (Table 3).

### 3.2. Modified Ashworth Scale and Penn Spasms Frequency Scale

Knee extensor and ankle planter flexors spasticity scores were significantly higher (*p* = 0.04) in group A compared to group B (Table 2). The Penn spams frequency scale showed more than 2× greater (*p* = 0.002) muscle spasms in group A compared to group B.

### 3.3. Muscle Quality

Figure 1 demonstrated the difference in absolute muscle CSA of the whole thigh and knee extensor muscles between group A and group B. Group A had a greater whole thigh muscle CSA in the right (98 ± 24 vs. 71 ± 30 cm^2^; 27%; *p* = 0.08) and left (102 ± 23 vs. 69.5 ± 26 cm^2^; 32%; *p* = 0.02) legs compared to group B. DXA-leg lean mass was greater in group A (right:13%, *p* = 0.19 and left: 15%; *p* = 0.12) compared to group B but did not attain statistical significance.

### 3.4. Isometric Knee Extensor Peak Torque

Knee extensor peak isometric torques were different at 10 Hz (18 ± 9 vs. 5.5 ± 3.8 Nm; *p* = 0.009), 30 Hz (32 ± 9 vs.12.3 ± 10 Nm; *p* = 0.002) and 100 Hz (35 ± 15 vs. 14.5 ± 10 Nm; *p* = 0.014) between group A and group B.

### 3.5. Bone Quality

Right knee distal femur BMD (Figure 2) showed a trend (*p* = 0.08) between both groups (A: 0.92 ± 0.16 vs. B: 0.70 ± 0.16 g/cm^2^; 24%). Right proximal tibia (Figure 2) was statistically different (*p* = 0.03) between both groups (A: 1.1 ± 0.18 vs. B: 0.76 ± 0.29 g/cm^2^; 29%). The left side, although was still greater in group A compared to group B, did not attain statistical significance for the distal femur BMD (*p =* 0.46; 33%) or proximal tibia BMD (37%; *p* = 0.39).

## 4. Discussion

The major findings of this investigation indicate that surface NMES can be successfully used as a screening tool to distinguish between those with acceptable versus unacceptable neuro-musculoskeletal qualities. Persons with greater musculoskeletal qualities have a lower motor threshold to evoke leg extension repetitions with and without ankle weights. This is demonstrated with greater muscle CSA and greater peak isometric torque, higher BMD at both proximal tibia and distal femur and as enhanced neuromuscular excitability, as shown by greater measures of spasticity, Penn spasm frequency scales and knee extensor isometric peak toque. The findings indirectly highlight the interplay between neuromuscular and musculoskeletal systems in persons with SCI.

The speculative mechanisms supporting these current findings are manifold. Persons with chronic SCI likely to experience spasticity and frequent muscle spasms throughout the day [32,33]. Spasticity has been previously shown to maintain muscle CSA and explained 55% of the variance in muscle size in persons with incomplete SCI (AIS B and C); potentially via release of IGF-1 [32,34]. Spasticity not only helps maintaining muscle mass but prevents against infiltration of IMF [32]. Repetitive spasticity and muscle spasms may both be considered as low dose exercise through the day that help maintaining muscle quality [35]. This is similar to our findings following applications of 12–16 weeks of NMES-RT in persons with chronic SCI [25,26]. In the current work, muscle quality, defined as greater muscle CSA with lower infiltration of IMF and greater isometric peak torque [12,24], can be enhanced muscle quality in association with greater muscle stiffness [10], which likely increases mechanical stress on proximal tibia and distal femur according to Wolfe’s law [36]. This proposed mechanism may explain the current findings of greater knee BMD in group A.

Several reports demonstrate that spasticity may play a predisposing protective role against muscle atrophy and bone loss in persons with SCI [32,37]. On the contrary, anti-spasticity medications, including high doses of oral baclofen, may trigger arrays of protein degradation signaling pathways that override the protective effects of spasticity on muscle and bone. This was demonstrated in a case report where long-term intrathecal baclofen resulted in remarkable demineralization of the metaphysis and epiphysis of the distal femur and proximal tibia in a person with SCI [38]. This investigation’s current findings shed the light on the mechanical interplay between muscle and bone [14], which may be mediated via increasing spasticity, muscle stiffness and exerting continuous, non-exercise, tension on the both the proximal tibia and distal femur.

For the current study, musculoskeletal quality was determined by a number of parameters previously evaluated in persons with SCI. For example, muscle quality had been defined based on the extent of infiltration of IMF [7,8], with greater %IMF infiltration resulting in greater impedance to the propagation of current amplitude [24]. Furthermore, increased %IMF decreases muscle stiffness in persons with SCI [10] and IMF may impair glucose tolerance and insulin resistance [7]. Surface NMES-RT has been shown to evoke remarkable hypertrophy of the paralyzed lower extremity muscle, decrease IMF, and improve glucose tolerance and insulin sensitivity in persons with SCI [25,26]. Ryan et al., showed that 16 weeks of NMES-RT resulted in a 25% increase in mitochondrial capacity as measured by the phosphocreatine recovery rate [39]. Importantly, NMES-RT may represent a potential rehabilitation tool for evoking muscle hypertrophy, reducing IMF and enhancing bone quality in persons with SCI [11,12,26].

Peak torque relative to muscle CSA is another potential definition of muscle quality. Holman and Gorgey showed that knee extensor muscle quality was enhanced by the addition of testosterone treatment to NMES-RT [11], where isometric peak torque increased by 48% and muscle CSA by 31%. They further demonstrated greater tibial trabecular thickness, increased plate width and decreased trabecular spacing following 16 weeks of training in persons with SCI [11]. In this trial, we noted greater peak isometric torque in group A and since persons with SCI suffer from deterioration in the trabecular bone microarchitecture and an associated increase the vulnerability of bone fracture [21,40], the greater peak isometric torque in response to NMES application may reflect greater muscle stiffness and higher BMD at the level of the knee joints.

Presently, the use of resource intensive (i.e., costly, highly utilized, in limited supply) imaging techniques are the most acceptable diagnostic tools to determine both muscle and bone qualities [5,6,7,8,9]. Clinical trials geared towards standing and stepping have demonstrated the needs of these imaging techniques as safety standards to evaluate bone health [1,2], and lower BMD is considered an exclusionary criterion for number of clinical trials. This has necessitated the use of DXA and computerized tomography to exclude risk of osteoporosis after SCI. Additionally, current guidelines recommend DXA scan every 12 months to ensure monitoring of bone health in persons with SCI [41]. Beyond the radiation hazards, both MRI and DXA scanning techniques involve specific level of training, are expensive when used longitudinally in clinical settings, and may be not readily available in many settings due to high usage or setting (e.g., rural). In a VA cooperative study, the ilunar DXA was used as a safety measure to evaluate bone health at the sites of the knees or the hips before conducting home-based exoskeleton intervention for 16 weeks [42]. and the researchers relied on establishing a cut-off of 0.6 g/cm^2^ at the knees and T-scores of less than −3.5 SD at the neck of the femur to either include or exclude participants from the trial [16,19]. These anatomical sites were primarily chosen because they are the most prominent sites for fracture in persons with SCI [13,23,38]. However, participants may erroneously excluded from these trials because of failure to sustain appropriate positioning (i.e., knee or hip contractures) on the DXA table, frequent muscle spasms in supine lying, or interfering rods or screws that prevent accurate evaluation of BMD. In these instances, the use of NMES may provide a simple and safe technique for clinicians and researchers to guide their endeavors to limit the number of patients undergoing imaging techniques and to identify those who are at risks of developing neurogenic osteoporosis which may lead to condylar fractures in the future.

The NMES amplitude outcomes demonstrated consistency over the course of three weeks of training, as determined by strong correlations from week to week. We hypothesized that adding ankle weights may have resulted in different outcomes; however, all participants who were originally assigned to greater or lower musculoskeletal qualities maintained their group assignments independent of ankle weights. Adding ankle weights is likely to change the geometry of the stimulated muscle as well as surrounding soft tissues, especially subcutaneous adipose tissue. This may necessitate an increase in the current amplitude required to evoke full leg extension. However, the current amplitude remained less than 100 mA in the high musculoskeletal quality group (group A) and greater than 100 mA in the lower musculoskeletal quality group (group B). To attempt to control for this, we relied on two criteria by including the number of repetitions of leg extension; in addition to lower current amplitude, leg extension repetitions exceeding 70 reps (2 sessions per week) per leg has to be maintained. This arbitrary choice of leg extension repetitions was based on anecdotal evidence that neuromuscular fatigue commonly ensues towards the beginning of the fourth sets of training (i.e., 30 reps per session or 60 reps per week). Therefore, the greater number of repetitions (>70 reps per week) may be used as a proxy index of lower muscle fatigue and greater muscle endurance capacity during NMES training. It is unclear whether those with different musculoskeletal qualities will show similar extent of muscle hypertrophy or will respond differently as result of longitudinal exercise training. We have just recently studied the factors that may elucidate different muscle hypertrophy response in persons with SCI [43].

### Limitations

There are number of limitations of the current findings including the small sample size and inclusion of two women in the study. In general recruitment of women with SCI is challenging based on the disproportionate sex distribution (4 men: 1 woman) of those with SCI. Additionally, female body composition is different than men with SCI, with the tendency to store more subcutaneous adipose tissue in the pelvic-femoral area, which is likely to alter the propagation of the current amplitude. Of note, the women were split evenly between both groups. The current study is also limited to the type of NMES unit, the type of adhesive electrodes and anatomical location of the electrodes. The Richmar unit provides current amplitude up to 200 mA in a biphasic pulse that was adjusted at 450 µs. The choice of 100 mA as an arbitrary point corresponds to 50% of the maximum current amplitude. However, most commercially available units may have current amplitude that also does not exceed 100 mA. Additionally, reliance on adhesive gel type electrodes may have ensured high conductivity and facilitated a reduction in the impedance of the current. In our practice, this adhesive gel electrodes may last 3–4 weeks per participant before they need to be replaced. Different types, shapes and size of electrodes are available, which may limit the generalizability of the current findings. Knee extensor muscles are markedly atrophied in persons with SCI; and therefore, large size rectangular electrodes with a wide pulse duration (450 µs) and long inter-distance between electrodes may increase the number of motor units recruited to evoke full leg extension. Lastly, the discrepancy in BMD between the right and left legs are attributed primarily to the small sample size in both groups.

## 5. Conclusions

Surface NMES amplitude and leg extension repetitions successfully differentiated between greater and lower musculoskeletal qualities in individuals with chronic SCI. Those with greater muscle and bone qualities required lower current amplitude (<100 mA) to elicit necessary motor pattern to evoke full leg extension, and they were capable of maintaining over 70 repetitions throughout the course of three week with and without ankle weights. MRI and DXA findings agreed with NMES cut-offs. Those with greater musculoskeletal qualities had greater absolute whole thigh and knee extensor muscle CSAs, higher peak isometric torque and greater BMD of the distal femur and proximal tibia. Spasticity and muscle spasms were predisposing factors that facilitated maintenance of muscle and bone qualities. The findings indirectly suggest that there is interplay exist between muscle and bones in persons with SCI. Therefore, based on these findings, we propose the use of NMES as a simple and inexpensive diagnostic tool to indirectly assess musculoskeletal quality in persons with SCI. Future research should focus on the efficacy of this method in larger trials.

## Figures and Tables

**Figure 1 jcm-11-06681-f001:**
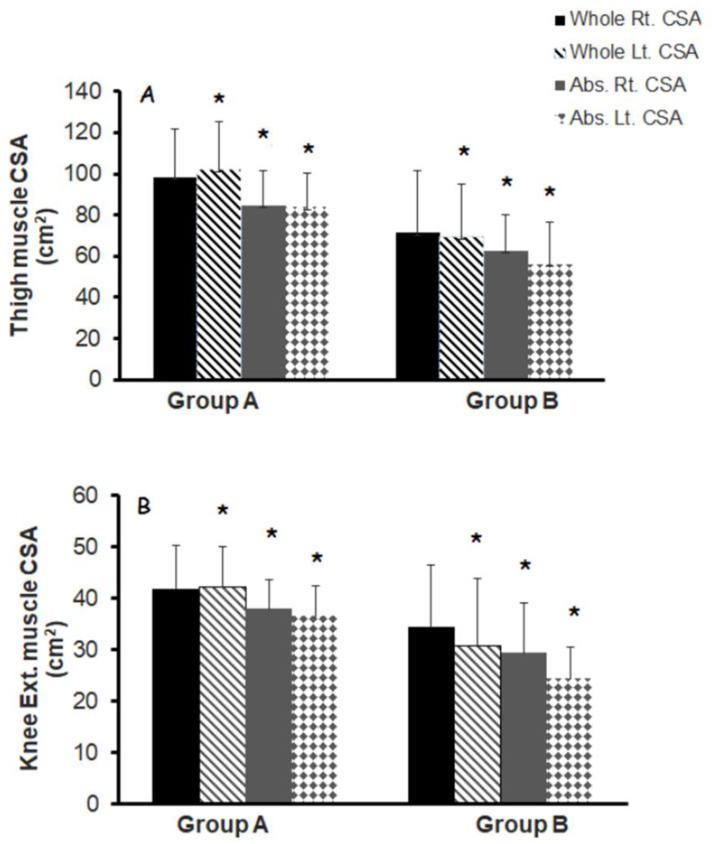
Whole thigh and knee extensor muscle CSAs with different musculoskeletal qualities in groups (**A**,**B**) classified based on NMES amplitude (greater or less than 100 mA) and leg extension repetitions (greater or less than 70 reps per week) in persons with chronic SCI. * denotes statistical significance between groups.

**Figure 2 jcm-11-06681-f002:**
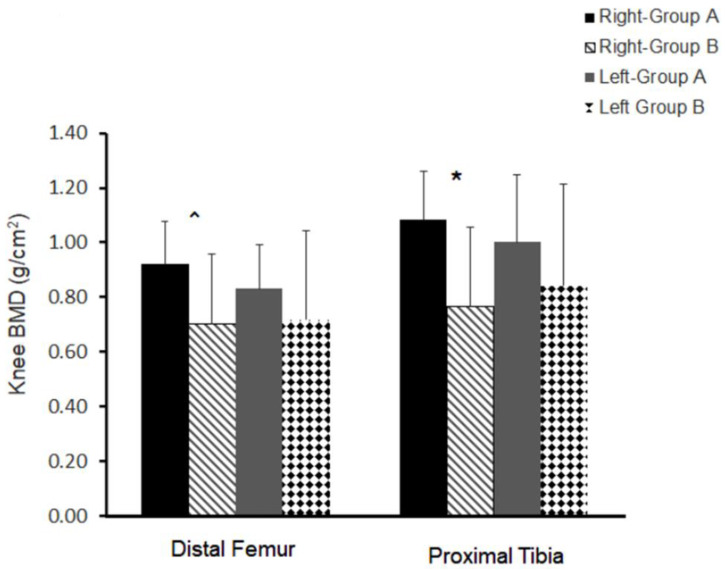
Distal femur and proximal tibia BMD with different musculoskeletal qualities in groups A and B classified based on NMES amplitude (greater or less than 100 mA) and leg extension repetitions (greater or less than 70 reps per week) in persons with chronic SCI. * denotes statistical significance between groups. ^ denotes trend between groups.

**Table 1 jcm-11-06681-t001:** Physical and SCI characteristics of the participants at the time of admission who were classified according to the amplitude of the current and repetitions into greater (group A) and lower (group B) musculoskeletal qualities. Values are presented as mean ± standard deviation (SD).

Characteristics	Group A (*n* = 8)	Group B (*n* = 9)	Between Group *p*-Values
Age (years)	38 ± 12	37 ± 13	0.88
Ethnicity	African American (*n* = 3) White (*n* = 5)	African American (*n* = 4) White (*n* = 5)	-
Gender	7 male, 1 female	8 male, 1 female	
Weight (kg)	74 ± 14	69 ± 21	0.51
Height (m)	1.75 ± 1.0	1.74 ± 0.7	0.7
BMI (kg/m^2^)	24.3 ± 5.4	22.7 ± 6.8	0.58
Paraplegia/Tetraplegia	7/1	4/5	-
Single Neurological Level (SNL)	C5-T12	C6-T12	-
TSI (years)	12.2 ± 11.0	12.0 ± 11.0	0.96
ISNCSCI classification	A (*n* = 2) B (*n* = 4)	A (*n* = 7) B (*n* = 1)	-
	C (*n* = 2)	C (*n* = 1)	

BMI: body mass index; ISNCSCI: International Standards for Neurological Classification of Spinal Cord Injury; TSI, time since injury; *n*, number.

**Table 2 jcm-11-06681-t002:** Modified Ashworth Scale (MAS), Penn spasm frequency scale, amplitude of NMES-current, for SCI groups with different musculoskeletal qualities. Values are presented as mean ± standard deviation (SD).

	Characteristics	Group A (*n* = 8)	Group B (*n* = 9)	*p*-Values
Hip MAS	Flexors	1 ± 1.07	0.77 ± 0.64	0.64
Knee MAS	Extensors Flexors	1 ± 1.1 1.12 ± 1.25	0.77 ± 0.97 0.44 ± 0.52	0.68 0.18
	Extensors	1.75 ± 1.0	0.77 ± 0.83	0.053
Ankle MAS	Dorsiflexors	0.75 ± 0.89	0.33 ± 0.50	0.26
	Planter flexors	1.12 ± 0.99	0.22 ± 0.44	0.04
Penn Spasm Frequency Scale		2.9 ± 0.83	1.1 ± 1.16	0.002
Amplitude of the current—Week 1 (mA)	Sets 1–2 (R/L) Set 3–4 (R/L)	73 ± 14/70 ± 15 80 ± 15/76 ± 20	137 ± 33/131 ± 33 148 ± 36/145 ± 32	0.0002/0.0008 0.0003/0.0002
Repetitions	Week 1 (R/L)	75 ± 14/75 ± 13	50 ± 22/55 ± 21	0.01/0.03
Weights (lbs.)	Week 1	0/0	0/0	
Week 2 (mA)	Sets 1–2 (R/L) Set 3–4 (R/L)	77 ± 17/70 ± 13 84 ± 19/77 ± 18	126 ± 37/125 ± 21 136 ± 34/137 ± 25	0.004/0.0003 0.001/0.0003
Repetitions	Week 2 (R/L)	80 ± 0/80 ± 0	56 ± 17/64 ± 24	0.002/0.07
Weights (lbs.)	Week 2	2 ± 0/1.8 ± 0.7	0.2 ± 0.7/0.3 ± 0.7	0.0001/0.0004
Week 3 (mA)	Sets 1–2 (R/L) Set 3–4 (R/L)	78 ± 18/77 ± 17 85 ± 22/85 ± 19	124 ± 33/123 ± 30 132 ± 33/134 ± 32	0.003/0.005 0.003/0.004
Repetitions	Week 3 (R/L)	70 ± 19/70 ± 16	61 ± 18/64 ± 21	0.3/0.4
Weights (lbs.)	Week 3	4 ± 0/3.2 ± 1.7	0.4 ± 0.8/0.9 ± 1.0	0.0001/0.0002

R: right leg; L: left leg. Both MAS and Penn spasm frequency scale were conducted prior to admission of the trial. 1lb is equal to 0.45 kg.

**Table 3 jcm-11-06681-t003:** Regression analysis models using body weight, amplitude of the current (mA) and repetitions as independent variables to predict muscle CSA, IMF CSA and BMDin persons with SCI.

	Dependent Variables	Independent Variables (Predictors)	R^2^	*p*-Values
Model 1	Whole thigh muscle CSA (cm^2^)	1.3 × body weight + 0.054 × amplitude of the current + 0.684 × reps-58.7	0.92	0.001
Model 2	Absolute whole thigh muscle CSA (cm^2^)	0.94 × body weight + 0.027 × amplitude of the current + 0.612 × reps-38.4	0.87	0.001
Model 3	Knee extensors muscle CSA (cm^2^)	0.499 × body weight + 0.045 × amplitude of the current + 0.301 × reps-23	0.88	0.001
Model 4	Absolute knee extensors muscle CSA (cm^2^)	0.401 × body weight + 0.031 × amplitude of the current + 0.276 × reps-17.3	0.84	0.001
Model 5	IMF-whole thigh CSA (cm^2^)	0.37 × body weight + 0.027 × amplitude of the current + 0.072 × reps-20.3	0.55	0.026
Model 6	Distal femur BMD (g/cm^2^)	0.006 × body weight + 0.002 × amplitude of the current + 0.006 × reps-0.248	0.67	0.016
Model 7	Proximal tibia BMD (g/cm^2^)	0.010 × body weight + 0.001 × amplitude of the current + 0.07 × reps-0.39	0.71	0.009

For simplicity purpose, all the seven regression models presented are for the right leg. Amplitude of the current (mA) is the average of the current in the first two sets of 10 repetitions in week 1. Repetitions are the total number of leg extension (40 reps per visit) achieved over two separate visits (40 reps × 2 visits = 80 reps) separated by 2–3 days in week 1.

## Data Availability

Data will be available upon direct contcat of the corresponding authors and oncer receive appropriate approvals for data sharing from the local ethical committee.

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
