# Peer review of "Employment of Neuromuscular Electrical Stimulation to Examine Muscle and Bone Qualities after Spinal Cord Injury"

_jcm, 2022, doi:10.3390/jcm11226681_

Round 1
Reviewer 1 Report
Dear authors, I have read the paper "Employment of Neuromuscular Electrical Stimulation to Examine Muscle and Bone Qualities after Spinal Cord Injury" and I find it interesting, with accurate discription of methods, results and discussion.
I have noticed that sometimes whole words are repeated and then acronyms, such as bone mineral density (BMD), and so I suggest the authors not repeat the whole word if they used abbreviations (line 262, 264).
The P values in Table 3 need to be realigned.
Author Response
Dear authors, I have read the paper "Employment of Neuromuscular Electrical Stimulation to Examine Muscle and Bone Qualities after Spinal Cord Injury" and I find it interesting, with accurate description of methods, results and discussion.
We would like to thank the reviewer for his/her time, feedback and comments about our submission.
I have noticed that sometimes whole words are repeated and then acronyms, such as bone mineral density (BMD), and so I suggest the authors not repeat the whole word if they used abbreviations (line 262, 264).
Thank you so much. This has been corrected throughout the manuscript.
The P values in Table 3 need to be realigned.
P-values were aligned in Table 3.

Reviewer 2 Report
The paper is interesting and well-written, just minor adjustments are necessary.
Major
None.
Minor
L. 14. The criteria for group division were not clear in the abstract.
L. 19. What are “CSAs”? The abbreviation was not defined in the abstract.
L. 126-128. “the current amplitude was gradually increased to achieve full knee extension. After achieving full knee extension, the current amplitude for each repetition was recorded.” this methodology was defined as the maximum electrically stimulated extension (MESE) in the paper: DOI 10.1007/s42600-020-00061-z.
L. 324 and others. Adjust the space between the word and citation, sometimes appearing “ SCI[33, 37].” and other times “SCI [38].”. Revise the manuscript.
Author Response
The paper is interesting and well-written, just minor adjustments are necessary.
We would like to thank the reviewer for his/her time, feedback and comments about our submission.
Major
None.
Minor
14. The criteria for group division were not clear in the abstract.
This was clarified in lines 13-15 “Participants were classified according to the current amplitude (> 100 mA) and the number of repetitions (> 70 reps) of leg extension into greater (n=8; 1 woman; group A) and lower (n=9; 1 woman; group B) musculoskeletal qualities”
19. What are “CSAs”? The abbreviation was not defined in the abstract.
Thank you. This was clarified.
126-128. “the current amplitude was gradually increased to achieve full knee extension. After achieving full knee extension, the current amplitude for each repetition was recorded.” this methodology was defined as the maximum electrically stimulated extension (MESE) in the paper: DOI 10.1007/s42600-020-00061-z.
Thank you so much for pointing to this important work. We have cited the work in our manuscript as reference # 27.
324 and others. Adjust the space between the word and citation, sometimes appearing “ SCI[33, 37].” and other times “SCI [38].”. Revise the manuscript.
Thank you so much. This was fixed throughout the manuscript.
